# Standardized effect sizes are far from "Standardized": A primer and empirical illustration in depression psychotherapy meta-analyses

Mathias Harrer[1,2]*, Clara Miguel[1], Yan Luo[3], Edoardo G. Ostinelli[4,5,6], Eirini Karyotaki[1], Stefan Leucht[2], Toshi A. Furukawa[7], Pim Cuijpers[1]

**1** Department of Clinical, Neuro and Developmental Psychology, World Health Organization Collaborating Center for Research and Dissemination of Psychological Interventions, Amsterdam Public Health Research Institute, Vrije Universiteit, Amsterdam, The Netherlands, **2** Section for Evidence-Based Medicine in Psychiatry and Psychotherapy, Department of Psychiatry and Psychotherapy, School of Medicine and Health, German Center for Mental Health (partner site Munich-Augsburg), Technical University of Munich, Munich, Germany, **3** Department of Health Promotion and Human Behavior, Kyoto University Graduate School of Medicine and School of Public Health, Kyoto, Japan, **4** Department of Psychiatry, University of Oxford, Oxford, United Kingdom, **5** Oxford Precision Psychiatry Lab, NIHR Oxford Health Biomedical Research Centre, Oxford, United Kingdom, **6** Oxford Health NHS Foundation Trust, Warneford Hospital, Oxford, United Kingdom, **7** Kyoto University Office of Institutional Advancement and Communications, Kyoto, Japan

* mathias.harrer@tum.de

## Abstract

Standardized mean differences (SMDs) are frequently used to appraise the effects of psychological treatments, and to combine them in meta-analyses. Yet, there is no consensus on how exactly SMDs should be computed from randomized trials. In this study, we show that different SMD variants can heavily diverge in aggregate-data meta-analyses, subverting the original purpose of standardization. We investigate the impact this has on the estimated benefits of depression psychotherapies. Different SMD versions using endpoint or change scores were calculated from a comprehensive database of randomized trials, comparing depression psychotherapy against pharmacotherapy and inactive controls. Pooled treatment effects were obtained for each variant, assuming correlations between baseline and endpoint scores of 0.2 through 0.8, and their relationship was examined using bivariate meta-analyses. We also investigated which study characteristics predicted divergent effect estimates. A total of $k = 443$ trials with 48,221 participants were analyzed. The pooled effect of psychotherapy versus controls varied heavily depending on the calculation methods (SMD = 0.65–1.24), even though the same studies were used. Divergences were less pronounced for psychotherapies compared to pharmacotherapy (SMD = 0.05–0.14). Change score SMDs deviated from endpoint SMDs especially when high ($r = 0.8$) or low ($r = 0.2$) pre-post correlations were assumed. This difference was largest in subfields with high treatment effects. Different SMD calculation methods can lead

**Data availability statement:** All code used for the analyses is openly available on Zenodo (https://doi.org/10.5281/zenodo.10694719).

**Funding:** There was no funding for the present work. Outside the present work, EGO was supported by the National Institute for Health and Care Research (NIHR) Research Professorship (grant RP-2017-08-ST2-006), by the National Institute for Health Research (NIHR) Applied Research Collaboration Oxford and Thames Valley (ARC OxTV) at Oxford Health NHS Foundation Trust, by the NIHR Oxford Health Clinical Research Facility, by the NIHR Oxford Health Biomedical Research Centre (grant BRC-1215-20005), and by the Brasenose College Senior Hulme scholarship. The views expressed are those of the authors and not necessarily those of the UK National Health Service, the NIHR, or the Department of Health and Social Care.

**Competing interests:** I have read the journal's policy and the authors of this manuscript have the following competing interests: MH is a part-time employee of Get.On Institut GmbH/HelloBetter, a company that implements digital mental health interventions into routine care. EGO received research and consultancy fees from Angelini Pharma. In the last three years, SL has received honoraria for advising/consulting and/or for lectures and/or for educational material from Angelini, Apsen, Boehringer Ingelheim, Eisai, Ekademia, GedeonRichter, Janssen, Karuna, Kynexis, Lundbeck, Medichem, Medscape, Mitsubishi, Neu-rotorium, Otsuka, NovoNordisk, Recordati, Rovi, Teva. All other authors have declared that no competing interests exist.

to strongly diverging effect estimates of psychological treatment; especially when change scores are used and pre-post correlations are very high or low. This could have a profound impact on how treatment benefits are interpreted within and across meta-analyses. Researchers could prioritize endpoint SMDs of randomized trials, and should consider standardization using population-level estimates to improve the comparability of meta-analytic effects in the field.

**Open Material; Registration:** https://doi.org/10.5281/zenodo.10694719; https://osf.io/yx5jg; https://osf.io/4j23t.

## Introduction

In mental health research, rating scales are widely used to assess patients' symptom severity. Measurements are typically obtained by creating a sum score of all evaluated items, a practice that is not uncontroversial [1]. Such scales are also common in randomized controlled trials (RCTs), where the instrument is administered in both the intervention and control group at one or several assessment points. Randomization ensures exchangeability of the average potential outcome values, meaning that the causal treatment effect is identifiable by comparing the expected sum score of both groups at the same point in time [2].

There are different methods to calculate this estimand in practice, typically resulting in an estimated mean difference between the two groups. This value can then be used to assess the size of the treatment effect. However, for most mental disorders (including depression [3]), several symptom inventories are available, and if trials employed different scales, their mean differences cannot be directly compared. This has led to a widespread adoption of "standardized" effect sizes, most notably the standardized mean difference (SMD; Cohen's *d*). To standardize the effect, the mean difference is divided by a standardizing denominator, usually the pooled standard deviation (SD) of the sample. This step is essential when trials using different rating scales are combined in meta-analysis, and often employed in the social sciences [4]. Even when the same scale was used, SMDs may sometimes be favored over unstandardized MDs. For instance, health professionals are more likely to interpret SMDs correctly [5,6], and SMDs may also offer somewhat greater generalizability compared to MDs (defined as lower cross-study variability not attributable to sampling error, and greater agreement in effect estimates across studies [7]).

While commonly used, this standardization is not without flaws. First, it introduces ambiguity concerning which pooled SD should be selected (i.e., baseline, change, or endpoint scores). Moreover, it creates additional variability because SDs are estimated from individual trials, which often have limited sample sizes, and may differ in how narrowly defined their patient population was [8,9]. Previous work indicates that SMD calculation methods in the literature are very heterogeneous, including in psychiatric research; and that effect estimates can vary dramatically within the same study [10].

If used in meta-analyses, SMDs can have a substantial impact on an entire research field, and they are often used to inform treatment guidelines. This warrants a closer look if different ways to calculate a "standardized" effect will indeed yield consistent results. To this end, we first review approaches to calculate treatment effects from an RCT, including methods often used in aggregate-data meta-analyses. Then, we derive how and when these methods predict differences in "standardized" expressions of the effect.

## SMD calculation methods: A primer

For RCTs with pre-test measure $X$ and post-test measure $Y$, a frequently used approach to estimate the average treatment effect $\theta$ is an analysis of covariance (ANCOVA). ANCOVA implies a linear model in which $Y$ is regressed on a treatment indicator $T$ and (one or multiple) baseline measures $X$. Here, we assume that $X$ is the pre-test measure of the (continuous) outcome, and the only variable to be controlled for in the model. This gives us the following conditional expectation for $Y$ [2,11]:

$$\mathbb{E}\left[Y_a \big| X_a, T\right] = \alpha + \theta T + \beta_a X_a. \tag{1}$$

In (1) above, $a$ represents the control or intervention group, with $T = I(a = \text{Treatment})$, and with $\theta$ being the average treatment effect; while $\beta_a$ quantifies the slope between the pre- and post-test scores. ANCOVA-type models are frequently recommended in the analysis of RCTs, since adjustment for the baseline scores controls for between-group differences in prognostically relevant variables (i.e., realized confounding [12,13]) and improves power [14–16].

Covariate adjustment typically requires that individual participant data (IPD) is available for the trial. However, in meta-analyses of aggregate data, only the group means or group-wise mean change from baseline may be reported instead. This means that meta-analysts may be forced to obtain effect estimates from change or endpoint scores only, without any further adjustment. Both approaches can be re-expressed as special cases of the ANCOVA model given in (1).

First, we assume that the difference in change scores between the two groups is used to determine the treatment effect. Following Laird [17], we can rearrange (1) so that:

$$\mathbb{E}\left[Y_a - X_a \big| X_a, T\right] = \alpha + \theta T + (\beta - 1)X_a$$

$$\mathbb{E}\left[Y_a - X_a \big| T\right] = \alpha + \theta T \quad \text{iff.} \quad \beta = 1 \tag{2}$$

This equation offers two insights. Firstly, it emphasizes that using change scores as the outcome will yield identical results to a standard ANCOVA if baseline scores are additionally controlled for. Secondly, it shows that a "crude" analysis of change scores without baseline adjustment will only be identical to ANCOVA when the slope between pre-test and post-test scores $\beta$ is exactly one (since $(\beta - 1)X_a$ will only drop out of the equation when $\beta = 1$). Sometimes, it is assumed that using change scores will control for baseline symptom severity; (2) above shows that this only holds when $\beta$ is exactly one, which is unlikely to occur in practice [11].

If an unadjusted analysis of the endpoint scores is used, (1) reduces to:

$$\mathbb{E}\left[Y_a \big| T\right] = \alpha + \theta T \quad \text{iff.} \quad \beta = 0 \tag{3}$$

Effectively, this approach ignores the relationship between pre- and post-test scores, thus setting $\beta = 0$. In RCTs, this approach remains asymptotically unbiased but is generally less efficient than ANCOVA with baseline score adjustment [13]. Notably, mean differences derived from both change scores (equation 2) and endpoint scores (equation 3) provide unbiased estimates of the treatment effect in successfully randomized trials, though their efficiency may vary [11].

A further complication in meta-analyses is that different instruments (e.g., depression scales) may have been used across studies, and that mean differences $\hat{\theta}$ estimated from each trial are therefore not comparable. This can be resolved by calculating a "unit-free" [18] measure of the effect, viz., the SMD. A generic definition of this standardized effect $\theta^*$ is [19]:

$$\theta^* = \frac{\theta}{\sigma} = \frac{\mu_1 - \mu_2}{\sigma} \tag{4}$$

where $\mu_1$ and $\mu_2$ are the (independent) population-level means of two populations, and $\sigma$ is the SD based on either population (where $\sigma = \sigma_1 = \sigma_2$). A practical problem is what empirical estimates ought to be plugged into equation (4). In an analysis of endpoint scores, the SMD is typically calculated using this formula [20]:

$$SMD_{EP} = \frac{\overline{Y}_{int} - \overline{Y}_{ctrl}}{SD_{EP}} J\left(n_{int} + n_{ctrl} - 2\right). \tag{4}$$

Where the second part of the formula applies a small-sample bias correction, with function $J(\cdot)$ defined as:

$$J(\nu) = \frac{\Gamma(\nu/2)}{\sqrt{(\nu/2)\Gamma((\nu-1)/2)}} \tag{5}$$

where $\Gamma$ is the gamma function and $\nu$ the degrees of freedom. This small sample bias corrected SMD is commonly known as Hedges' $g$. The standardizing denominator $SD_{EP}$ in (4) is the pooled SD of the endpoint $Y$ in both groups:

$$SD_{EP} = \sqrt{\frac{(n_{int}-1)SD^2_{Y,int} + (n_{ctrl}-1)SD^2_{Y,ctrl}}{n_{int} + n_{ctrl} - 2}} \tag{7}$$

The SMD based on change scores can be calculated in a similar manner:

$$SMD_{CS} = \frac{(\overline{Y}_{int} - \overline{X}_{int}) - (\overline{Y}_{ctrl} - \overline{X}_{ctrl})}{SD_{CS}} J\left(n_{int} + n_{ctrl} - 2\right). \tag{8}$$

However, since $SMD_{CS}$ makes use of both $Y$ and $X$, the standardizing denominator is less clearly defined. Some propose that the pooled pre-test SD should be used [21,22]; while others define $SD_{CS}$ as the SD of the change scores [11,23]:

$$SD_{CS,a} = \sqrt{SD^2_{Y_a} + SD^2_{X_a} - 2r_a SD_{Y_a} SD_{X_a}} \tag{9}$$

Where $a$ is the intervention or control group, and $r_a$ is the in-sample correlation coefficient between the pre- and post-test scores. A practical problem with (9) above is that such trial-specific correlation coefficients are rarely reported; the value of $SMD_{CS}$ obtained using this method will therefore heavily depend on the value imputed for $r_a$. In meta-analyses, $r_a$ values can be imputed using representative values from the literature, or sourced from other studies in the meta-analysis that provide empirical estimates. In some cases, $r_a$ may also be approximated from other reported summary statistics [24].

Senn [11,25] shows that, under some simplifying assumptions, the different analytical strategies (ANCOVA, analysis of endpoint scores, analysis of change scores) are strictly related. Given equal variances of the baseline and endpoint scores ($\sigma^2_{Y_a} = \sigma^2_{X_a}$), as well as equal pre-post correlations $\rho$ and sample sizes in both groups, we obtain the following equality for effect estimates of the three approaches:

$$\hat{\theta}_{ANCOVA} = \rho\hat{\theta}_{CS} + (1-\rho)\hat{\theta}_{EP}. \tag{10}$$

This shows that effect estimates based on change scores ($\hat{\theta}_{CS}$) will be closer to the ANCOVA estimate when the pre-post correlation is high (i.e., $\rho > 0.5$). If the correlation is lower ($\rho < 0.5$), the unadjusted endpoint estimate ($\hat{\theta}_{EP}$) will be closer.

Under these assumptions, we can also directly define the relationship between the SD estimates that are used to "standardize" the mean difference in the denominator:

$$SD^2_{EP} = \frac{SD^2_{CS}}{2(1-\rho)} = \frac{SD^2_{ANCOVA}}{1-\rho^2} \tag{11}$$

This equation again underlines the importance of the pre-post correlation: for large correlations (i.e., $\rho > 0.5$), $SD_{EP}$ will be larger than $SD_{CS}$; thus, given the same estimated mean difference, $SMD_{CS} > SMD_{EP}$. This reverses for smaller correlations ($\rho < 0.5$): $s_{EP}$ is now smaller than $SD_{CS}$, so that $SMD_{CS} < SMD_{EP}$. Importantly, (11) above also shows that, in almost all contexts, the SD of the ANCOVA model will be smaller than the one based on change scores or endpoints only (since $1 - \rho^2 \leq 2(1-\rho)$ and $1 - \rho^2 \leq 1$). This relationship also translates to the sampling variances $Var[\hat{\theta}_{EP}]$, $Var[\hat{\theta}_{CS}]$, and $Var[\hat{\theta}_{ANCOVA}]$ [11]. In sum, this shows that different SMD calculation methods can produce widely varying results, even when the true treatment effect is the same. These discrepancies primarily stem from the choice of standardizing denominator. In S1 Text, we provide a visual summary of these predicted differences as obtained by a simulation study.

The same definitions as shown above are also given in an influential treatment by Cohen [26]. However, Cohen discusses these different ways to obtain the standardizing denominator in the context of power analyses. In this setting, it is clearly sensible to adapt the standardizing divisor to the analytic approach to be used in the study: given the same sample size and mean difference, adjusting for baseline will yield higher power estimates because it decreases $SD$, thus yielding a higher SMD to be considered in the power analysis. Adjusting for covariates in an ANCOVA will almost always increase the statistical power compared to a crude analysis of endpoint means; for change scores, this will only be the case if the pre-post correlation is high.

It is questionable if this rationale translates well into the context of meta-analyses. Different ways to obtain the standardizing denominator mean that effect estimates will diverge depending on the method that was used to calculate the SMD (change score or endpoints), and the pre-post correlation that meta-analysts are willing to assume. Depending on what approaches are used, this may heavily limit the comparability of effect sizes within and across meta-analyses. Researchers could seriously over- or underestimate the efficacy of a treatment if SMD estimates are compared to other trials or meta-analyses using a different standardization method, or if wrong assumptions about the pre-post correlation are made. Results of our "toy" simulation shown in S1 Text further illustrate this issue.

Cohen himself remarked on the limited transportability of standardized effect sizes [27,28]. SMDs and similar measures create a dependency between the effect size and the variability of a specific sample; this means that two identical patients with the same causal treatment benefits (e.g., a 5-point decrease on the PHQ-9 compared to not receiving treatment) would be judged to have experienced drastically different treatment effects if one was part of group that varies greatly, and the other part of a group with hardly any variation. This issue also extends to the different standardizing denominators we mentioned above: given the same aggregate data, the size of a treatment effect entered into meta-analysis will depend on the method used to obtain the SMD, and how efficient this approach is in the specific context of the study. Such a context-dependent divergence of identical causal treatment effects is clearly undesirable.

## Aims of the current study

It is important to note that the strict relationships between different calculation methods described in (10) and (11) are based on several simplifying assumptions (homogeneity of $\sigma^2_{Y_a}$ and $\sigma^2_{X_a}$; equal correlations and sample sizes in both groups). This is unlikely to hold in practice. More generally, it is uncertain what the real impact of these divergences will be in fields such as meta-analytic psychotherapy research, where SMDs are commonly used; and which types of studies

and treatments are most affected. A previous meta-epidemiological study indicates that SMD estimates can vary strongly within the same trial, especially in studies with small sample sizes and high treatment effects; but no approach appeared to produce consistently smaller or higher values [10]. In this study, we therefore aim to systematically investigate the impact of different SMD calculation methods on the estimated meta-analytic effect of psychotherapy for depression. Focusing on aggregate-data information reported in the publications, we will examine different approaches to obtain the mean difference between groups (endpoint scores versus change scores), as well as the standardizing denominator (pooled pre-test, change score, or endpoint SD), and examine how strongly these SMD variants can diverge from each other. We will also assess the influence of pre-post correlations (ranging from low to high) that meta-analysts may be willing to assume when calculating the SMDs.

## Method

A preregistration of our investigation has been published with the Open Science Framework (osf.io/4j23t). All code used for the analyses is openly available on Zenodo (doi.org/10.5281/zenodo.10694719).

### Datasets

Our analyses are based on two meta-analytic databases included in the "Metapsy" meta-analytic research domain (MARD) for psychological treatments [29,30] (metapsy.org). The Metapsy MARD provides comprehensive living databases of randomized trials for various indications and treatments, which are harmonized using a unified protocol [31]. The Metapsy databases have been used for more than 100 meta-analytic reviews published within the last 15 years [30,32] (see metapsy.org/published-articles for an overview). The exact search strategy, data extraction and coding for each database is detailed in the documentation page of the initiative (docs.metapsy.org/databases).

This study focuses on the "Depression: Psychotherapy vs. Control" (docs.metapsy.org/ databases/depression-psyctr) and "Depression: Psychotherapy vs. Pharmacotherapy" datasets, which are compiled using the same methodology [33]. The most recent update of the databases was used, including studies published until May 1st, 2023. Both database versions can be downloaded online (doi.org/10.5281/zenodo.15584092; "data" folder). In both databases, risk of bias is rated using four criteria of the "Risk of bias" (RoB) assessment tool, version 1, developed by Cochrane [34]. Assessed domains include the adequate generation of allocation sequence; the concealment of allocation to conditions; the prevention of knowledge of the allocated intervention (masking of assessors); and dealing with incomplete outcome data (this was assessed as positive when intention-to-treat analyses were conducted, meaning that all randomized patients were included in the analyses). Trials are judged as having a low risk of bias when they score positive on all four domains. Psychological treatments are categorized into one of eight types based on a pre-specified rationale [35]. Extractions also include group-wise attrition, defined as the number of participants who were lost to follow-up.

For both datasets, results of all depression symptom instruments are extracted from each trial. A pre-specified hierarchy is used when extracting the effect size data, giving priority to the raw mean, SD and sample size of each condition at baseline and follow-up.

Because our analysis focused on comparing different SMD calculation methods with each other, we only considered studies for which the arm-specific mean, SD and sample size at baseline and endpoint were available. We excluded studies that reported the mean change scores and their standard deviation directly, but not the means, SDs or sample sizes of scores at baseline and the endpoint, because some SMD variants cannot be not directly calculated from them. In the "Depression: Psychotherapy vs. Pharmacotherapy" dataset, we additionally excluded trial arms that did not employ psychotherapy or ADM as a monotherapy (e.g., combined therapy, psychotherapy and pill placebo, ADM and pill placebo).

## Calculation of change scores

Arm-specific mean change scores ($m_{CS}$) in the eligible trials were calculated by subtracting the mean depressive symptoms score at baseline from the endpoint mean ($m_{EP} - m_{BL}$). We assumed that sample sizes for the change score ($n_{CS}$) were identical to the sample size available at the endpoint ($n_{EP}$). We also calculated the SD of the change scores ($SD_{CS}$), using the formula given in (9), and assuming different pre-post correlations (see below).

## Calculation of SMDs

In all included studies, we first calculated the SMD using endpoints means, which were standardized by the pooled SD of the endpoint scores ($SMD_{EP/EP}$). Then, we also calculated three SMD variants based on change scores, dividing by either the (i) pooled pre-test SD ($SMD_{CS/BL}$), (ii) pooled change score SD ($SMD_{CS/CS}$), or (iii) pooled post-test SD ($SMD_{CS/EP}$). $SMD_{EP/EP}$, $SMD_{CS/BL}$, $SMD_{CS/CS}$ and $SMD_{CS/EP}$ represent distinct estimators, differing in both their numerator (endpoint vs. change scores) and denominator (baseline, endpoint, or change score SD), which may lead to substantial variations in the numeric value of the resulting effect size. All SMD versions were adjusted for small-sample bias using the correction factor described in equations (4) and (5).

The sampling variation $V$ of $SMD_{EP/EP}$ was calculated via the unbiased estimator given by Viechtbauer [28]:

$$V_{SMD_{EP/EP}} = \frac{n_{int} + n_{ctrl}}{n_{int} n_{ctrl}} + \left(1 - \frac{(m-2)}{mJ(m)^2}\right) SMD_{EP/EP}^2 \tag{12}$$

with $m = n_{int} + n_{ctrl} - 2$. The following delta method approximation was used for the sampling variances of $SMD_{CS/CS}$ and $SMD_{CS/EP}$:

$$V_{SMD_{CS/CS}} = J(m)^2 \left(\frac{2(1-r)(n_{int} + n_{ctrl})}{n_{int} n_{ctrl}} + \frac{SMD_{CS/CS}^2}{2m}\right)$$

$$V_{SMD_{CS/EP}} = J(m)^2 \left(\frac{2(1-r)(n_{int} + n_{ctrl})}{n_{int} n_{ctrl}} + \frac{SMD_{CS/EP}^2}{2m}\right)$$

$$\tag{13}$$

For $SMD_{CS/BL}$, we used the formula derived by Morris [21]:

$$V_{SMD_{CS/BL}} = 2\left(J(m)^2\right)(1-r)\left(\frac{n_{int} + n_{ctrl}}{n_{int} n_{ctrl}}\right)\left(\frac{m}{m-2}\right)$$

$$\left(1 + \frac{n_{int} n_{ctrl}}{n_{int} + n_{ctrl}} \frac{SMD_{CS/BL}^2}{2(1-r)}\right) - SMD_{CS/BL}^2. \tag{14}$$

Calculation of SMD versions based on change scores requires the value of the pre-post correlation to be imputed. In this analysis, we considered a range of possible correlation values $r \in (0.2, 0.4, 0.6, 0.8)$ [36], leading to a total of $3 \times 4 = 12$ variants of $SMD_{CS}$ being calculated for each comparison.

## Meta-analysis

For each of the SMD variants, we calculated the pooled effect of psychotherapy versus control groups, and of psychotherapy versus ADM, on depressive symptom severity. Different pooling models were considered: (i) a three-level "correlated and hierarchical effects" (CHE) model, assuming a constant sampling correlation of $\rho=0.6$ for effect sizes clustered within studies [37]; (ii) a generic inverse-variance random-effects pooling model, for which multiple effect sizes within studies were pre-aggregated to avoid double-counting (again assuming $\rho=0.6$); (iii) the same model as in (ii), but only using the highest or lowest effect size within a study; and (iv) the same model as in (ii), but excluding outliers and influential cases determined using the "leave-one-out" diagnostics by Viechtbauer and Cheung [38] (employing the same "rules of thumb" for outlier identification as used in the *influence.rma* function in *metafor* [39]). The restricted maximum likelihood (REML [40]) estimator was used to calculate the heterogeneity variance (components) $\tau^2$. The Knapp-Hartung adjustment was applied to the pooled effect in models (ii) to (iv) [41]. For model (i), cluster-robust variance estimation (CR2 estimator [42]) was used instead.

To examine the relationship between $SMD_{EP/EP}$ and the different $SMD_{CS}$ variants, we re-used the pre-aggregated effect estimates obtained for models (ii) to (iv) to perform a bivariate meta-analysis [43]. This model allows each trial to contribute two effect estimates, $SMD_{EP/EP}$ and one $SMD_{CS}$ variant, the true values of which are likely to be correlated. An unstructured heterogeneity variance-covariance matrix was used in the model, which allows the covariance between true effect sizes based on $SMD_{EP/EP}$ and $SMD_{CS}$ to be estimated across studies. We then used the results to regress the estimated true effects based on $SMD_{EP/EP}$ on the ones using the $SMD_{CS}$ variant. Ideally, $SMD_{EP}$ and $SMD_{CS}$ should not diverge, meaning that the estimated slope in this model should be exactly one. Thus, we also tested if the slope deviated significantly from this value. Bivariate models were fitted for each combination of $SMD_{EP/EP}$ and the $SMD_{CS}$ variants, and for all correlation values assumed in the effect size calculation step (i.e., $r=0.2$, 0.4, 0.6, and 0.8). To facilitate computations, while modelling the correlation of $SMD_{EP/EP}$ and $SMD_{CS}$ across trials, variances of the two SMD estimates were treated as conditionally independent within the same trial.

In a last step, we extended the bivariate models to examine if effect size divergences are moderated by study-level covariates. Examined moderators were (i) attrition (pooled across both groups); (ii) baseline imbalance (defined as the absolute value of the between-group SMD at baseline); (iii) the number of domains assessed to have a low risk of bias (range: 0–4); (iv) the type of control group; and (v) the type of psychological treatment used in the trial. This analysis was restricted to the "Depression: Psychotherapy vs. Control" database, for which a substantially larger number of studies was available.

All analyses were conducted in R version 4.2.0, using the *metapsyTools* package [44]. This extension imports functionality from the *meta* [45], *metafor* [39], *dmetar* [46] and *clubSandwich* [47] packages.

## Results

A total of $k=532$ trials were available in the two databases (psychotherapy versus control: $k=466$; psychotherapy versus ADM: $k=66$). After removing studies without reported pre- and post-test means, SDs or sample sizes, $k=443$ (83.3%) RCTs could be included in the analysis (psychotherapy versus control: $k=395$, 84.8%; psychotherapy versus ADM: $k=48$, 72.7%). In total, these trials enrolled 48,221 patients (psychotherapy versus control: 40,871; versus ADM: 7,350) and reported 902 effect measures (psychotherapy versus control: 791; versus ADM: 111). References for all included studies are provided in S2 Text. Overall, 156 (35.2%) trials met all four criteria for low risk of bias (psychotherapy versus control: 145, 36.7%; psychotherapy versus ADM: 11, 22.9%).

Pooled effects using the different SMD calculation methods are provided in Table 1. This table only shows results for the three-level CHE model; S1 Table and S2 Table give the results for all pooling models. Compared to the endpoint SMD estimate ($SMD_{EP/EP}=0.78$), effects of psychotherapy versus control groups were considerably higher when change score SMDs using baseline SDs were employed ($SMD_{CS/BL}=0.92$ to 0.93). Only small a difference emerged when change

**Table 1. Pooled effects of depression psychotherapy for different calculation methods of the SMD.**

| ρ | Calculation Method (SMD) | SMD | S.E. | 95% CI | $I^2$ | $\tau^2_{between}$ | $\tau^2_{within}$ | $\tau^2_{total}$ | 95% PI | NNT |
|---|---|---|---|---|---|---|---|---|---|---|
| **Psychotherapy vs. Control (k=395; 791 effect sizes)** | | | | | | | | | | |
| | SMD$_{EP/EP}$ | 0.78 | 0.038 | [0.71; 0.86] | 90.1 | 0.387 | 0.100 | 0.487 | [-0.59; 2.16] | 3.61 |
| 0.2 | SMD$_{CS/BL}$ | 0.92 | 0.041 | [0.84; 1.00] | 86.7 | 0.428 | 0.090 | 0.518 | [-0.55; 2.39] | 3.02 |
| | SMD$_{CS/CS}$ | 0.65 | 0.028 | [0.60; 0.71] | 76.4 | 0.222 | 0.037 | 0.259 | [-0.35; 1.65] | 4.43 |
| | SMD$_{CS/EP}$ | 0.82 | 0.038 | [0.75; 0.90] | 86.4 | 0.423 | 0.135 | 0.558 | [-0.59; 2.24] | 3.42 |
| 0.4 | SMD$_{CS/BL}$ | 0.92 | 0.041 | [0.84; 1.00] | 90.0 | 0.446 | 0.101 | 0.546 | [-0.58; 2.43] | 3.01 |
| | SMD$_{CS/CS}$ | 0.76 | 0.033 | [0.69; 0.82] | 86.4 | 0.326 | 0.065 | 0.391 | [-0.47; 1.98] | 3.76 |
| | SMD$_{CS/EP}$ | 0.83 | 0.038 | [0.75; 0.90] | 89.8 | 0.440 | 0.146 | 0.585 | [-0.63; 2.28] | 3.40 |
| 0.6 | SMD$_{CS/BL}$ | 0.93 | 0.038 | [0.85; 1.00] | 93.1 | 0.463 | 0.114 | 0.577 | [-0.61; 2.47] | 3.00 |
| | SMD$_{CS/CS}$ | 0.92 | 0.041 | [0.84; 1.00] | 93.5 | 0.506 | 0.122 | 0.628 | [-0.64; 2.47] | 3.04 |
| | SMD$_{CS/EP}$ | 0.83 | 0.041 | [0.75; 0.91] | 93.1 | 0.457 | 0.158 | 0.615 | [-0.66; 2.32] | 3.39 |
| 0.8 | SMD$_{CS/BL}$ | 0.93 | 0.041 | [0.85; 1.01] | 96.3 | 0.479 | 0.131 | 0.609 | [-0.65; 2.51] | 2.99 |
| | SMD$_{CS/CS}$ | 1.24 | 0.056 | [1.13; 1.35] | 97.9 | 0.953 | 0.268 | 1.221 | [-0.93; 3.41] | 2.20 |
| | SMD$_{CS/EP}$ | 0.83 | 0.038 | [0.76; 0.91] | 96.4 | 0.474 | 0.173 | 0.647 | [-0.70; 2.37] | 3.37 |
| **Psychotherapy vs. Antidepressive Medication (k=48; 111 effect sizes)** | | | | | | | | | | |
| | SMD$_{EP/EP}$ | 0.13 | 0.087 | [-0.04; 0.30] | 84.9 | 0.291 | 0.000 | 0.291 | [-0.95; 1.21] | 25.99 |
| 0.2 | SMD$_{CS/BL}$ | 0.05 | 0.107 | [-0.15; 0.26] | 82.3 | 0.330 | 0.025 | 0.389 | [-1.20; 1.31] | 64.47 |
| | SMD$_{CS/CS}$ | 0.05 | 0.082 | [-0.10; 0.21] | 69.3 | 0.181 | 0.000 | 0.181 | [-0.80; 0.91] | 64.77 |
| | SMD$_{CS/EP}$ | 0.09 | 0.097 | [-0.10; 0.28] | 80.4 | 0.364 | 0.000 | 0.330 | [-1.07; 1.24] | 40.23 |
| 0.4 | SMD$_{CS/BL}$ | 0.06 | 0.102 | [-0.14; 0.26] | 86.9 | 0.347 | 0.041 | 0.418 | [-1.24; 1.35] | 60.09 |
| | SMD$_{CS/CS}$ | 0.07 | 0.087 | [-0.10; 0.24] | 82.0 | 0.275 | 0.002 | 0.277 | [-0.99; 1.13] | 49.04 |
| | SMD$_{CS/EP}$ | 0.09 | 0.097 | [-0.10; 0.28] | 85.2 | 0.377 | 0.003 | 0.350 | [-1.10; 1.28] | 38.50 |
| 0.6 | SMD$_{CS/BL}$ | 0.06 | 0.107 | [-0.14; 0.27] | 91.4 | 0.361 | 0.059 | 0.453 | [-1.29; 1.41] | 55.76 |
| | SMD$_{CS/CS}$ | 0.10 | 0.102 | [-0.11; 0.30] | 91.8 | 0.431 | 0.027 | 0.458 | [-1.26; 1.45] | 36.12 |
| | SMD$_{CS/EP}$ | 0.09 | 0.097 | [-0.10; 0.28] | 90.1 | 0.394 | 0.012 | 0.374 | [-1.13; 1.32] | 36.80 |
| 0.8 | SMD$_{CS/BL}$ | 0.07 | 0.102 | [-0.14; 0.27] | 95.8 | 0.375 | 0.081 | 0.496 | [-1.34; 1.48] | 51.50 |
| | SMD$_{CS/CS}$ | 0.14 | 0.143 | [-0.14; 0.42] | 97.7 | 0.835 | 0.088 | 0.922 | [-1.78; 2.06] | 23.84 |
| | SMD$_{CS/EP}$ | 0.10 | 0.097 | [-0.09; 0.29] | 95.1 | 0.415 | 0.026 | 0.402 | [-1.17; 1.37] | 35.07 |

*Note.* Estimates are based on a three-level correlated and hierarchical effects model (CHE), assuming a constant sampling correlation of $\rho=0.6$ within studies. Results for other pooling models are presented in S1 Table and S2 Table. NNT = Number Needed to Treat; PI = Prediction Interval; SMD$_{EP/EP}$ = SMD calculated by dividing the mean endpoint difference by the pooled endpoint SD; SMD$_{CS/BL}$ = SMD calculated by dividing the mean change score difference by the pooled baseline SD; SMD$_{CS/CS}$ = SMD calculated by dividing the mean change score difference by the pooled change score SD; SMD$_{CS/EP}$ = SMD calculated by dividing the mean change score difference by the pooled endpoint SD.

score SMDs were standardized by the post-test SD ($SMD_{CS/EP}$=0.82 to 0.83). For change score SMDs standardized by the change score, divergences heavily depended on the assumed correlation. We found much higher effect estimates for $r$=0.8 ($SMD_{CS/CS}$=1.24, vs. $SMD_{EP/EP}$=0.78) and $r$=0.6 ($SMD_{CS/CS}$=0.92); but comparable effects when $r$=0.4 ($SMD_{CS/CS}$=0.76), and slightly lower results for $r$=0.2 ($SMD_{CS/CS}$=0.65). The estimated total heterogeneity variance mirrored this pattern, leading to higher values ($\tau^2_{total}$=1.221, vs. 0.487 for $SMD_{EP/EP}$) when assuming $r$=0.8, and to lower values ($\tau^2_{total}$ = 0.259) for $r$=0.2. Overall, the proportion of variation not attributable to sampling error was very large for all meta-analyses ($I^2$=76.4% to 97.9%), even when outliers and influential cases were removed ($I^2$=60.2% to 94.3%; see S1 Table).

Relationships between endpoint and change score SMDs as estimated using bivariate meta-analysis are visualized in Fig 1. For $SMD_{CS/BL}$, estimated slopes ranged from $\hat{\beta}$=1.160 to 1.197, and differed significantly from one (all $p$<0.05).

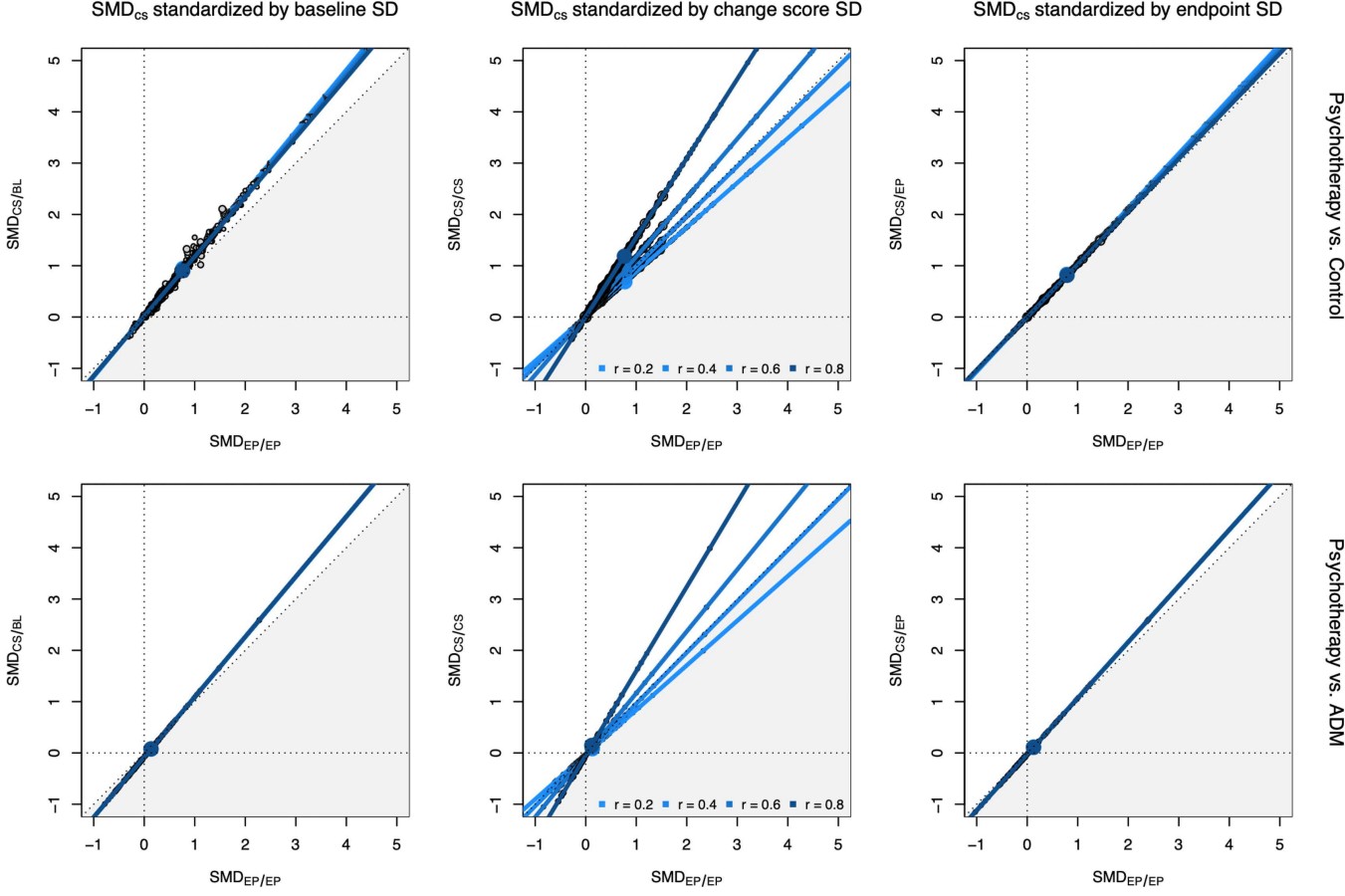

**Fig 1. *Estimated relationship between true effect sizes for different calculation methods of the SMD*.**

For $SMD_{CS/CS}$, the estimated slopes were $\hat{\beta}$=0.868 ($r$=0.2), 0.973 ($r$=0.4), 1.150 ($r$=0.6), and 1.544 ($r$=0.8), all diverging significantly from unity. For $SMD_{CS/EP}$, the estimated slopes ranged from $\hat{\beta}$=1.027 to 1.063 (all $p$<0.05).

Pooled effects of psychotherapy versus ADM were more comparable across calculation methods ($SMD$=0.05 to 0.14). However, for $SMD_{CS/CS}$, estimates of the total heterogeneity again heavily depended on the chosen correlation ($\tau^2_{total}$ = 0.181 to 0.922), with higher imputed correlations producing higher heterogeneity estimates. The percentage of variation not attributable to sampling error was also large ($I^2$=69.3% to 97.7%), and remained substantial even after outlier and influential case removal ($I^2$=37.3% to 91.9%; see S2 Table). Effect divergences for this dataset are provided in the bottom row of Fig 1. Despite the lower overall effect, the estimated slopes again displayed a similar pattern. For $SMD_{CS/BL}$, all slopes were significantly larger than one ($\hat{\beta}$=1.171 to 1.173, all $p$<0.05). Divergences of $SMD_{CS/CS}$ again varied heavily depending on the chosen correlation ($\hat{\beta}$=0.870 to 1.643); for $r$=0.4, true effects correlated almost perfectly ($\hat{\beta}$=0.997). For $SMD_{CS/BL}$, slopes were slightly larger than one ($\hat{\beta}$=1.093 to 1.094; all $p$<0.05).

S1 Fig shows differences between SMD calculation methods conditional on study and treatment characteristics. Results were consistent with the overall pattern established via bivariate meta-analysis. First, divergences were greatest for $SMD_{CS/CS}$ when high correlation values (i.e., $r$=0.8) were chosen ($\Delta_{SMD}$=0.35 to 0.52). Results for $SMD_{CS/BL}$ were less pronounced, but still led to significant effect differences for all comparators, and for some treatments (behavioral activation, cognitive behavior therapy, interpersonal therapy, problem solving therapy; $\Delta_{SMD}$=0.12 to 0.28). Across moderators,

**Table 2. Practical recommendations to avoid common pitfalls in effect size standardization.**

| Recommendation | Explanation |
|---|---|
| *Prioritize endpoint SMDs.* | We found that $SMD_{EP/EP}$ (see equations 4 and 7) can be calculated for nearly all included trials. For endpoint SMDs, the standardizing denominator is clearly defined as the pooled endpoint SD, and does not require the pre-post correlation in the trial to be known. Prioritizing $SMD_{EP/EP}$ could therefore be a suitable way to increase the comparability of estimated treatment effects across aggregate data meta-analyses or other evidence syntheses. |
| *If change score SMDs are included, minimize the reliance on (imputed) pre-post correlations.* | Change score SMDs can diverge significantly from other SMD variants when change score SDs are used in the denominator, as they depend heavily on the (imputed) pre-post correlation in both groups. We found that using endpoint SDs instead (yielding $SMD_{CS/EP}$) reduces discrepancies with endpoint SMDs. If endpoint SDs are not reported, they can be imputed from other studies [51]. |
| *Pre-specify and report the chosen SMD calculation method.* | Regardless of which SMD variant is used, ensure that the calculation method is (1) pre-specified in the protocol and (2) clearly described in the final report. This prevents selective reporting of the variant yielding the highest numerical estimate, and helps to trace back differences in pooled effects across similar meta-analyses. |
| *Consider using population-level standardizers.* | When calculating SMDs, standard deviations reported in the trial are conventionally used to approximate the population SD. Yet these study-specific SDs are only estimates, and may differ depending on the overall variability in the trial. To mitigate this issue, meta-analysts could use external reference SDs across studies using the same scale. If universally accepted reference SDs exist or can be established, they could greatly enhance the comparability of meta-analytic effect estimates. |
| *Promote core outcome sets.* | Many challenges in effect standardization could be avoided if all trials used the same measurement instrument. Core Outcome Sets (COS; [53]) have been developed for many subfields in mental health research, and their consistent implementation in new trials should be encouraged. |

*Note*. These are pragmatic recommendations for aggregate data meta-analyses of randomized trials in mental health research. Some of the points mentioned above may not translate to other fields, for example quasi-experimental designs. Please also note these are not recommendations for primary analyses of randomized trials, in which individual participant data is typically available. Suitable guidance on the primary evaluation of clinical trials in mental health research is provided elsewhere [2].

differences between $SMD_{CS/EP}$ and $SMD_{EP/EP}$ were mostly small and not significant. Second, we found that effect divergences were generally higher among subgroups of studies that generally produce high effect estimates. For $SMD_{CS/BL}$ ($r = 0.2$ to $0.6$) and $SMD_{CS/CS}$ ($r = 0.6$ and $0.8$), we found a significant moderating effect of study quality, whereby distances to $SMD_{EP/EP}$ increase with the number of domains judged to have a high or unclear risk of bias. Baseline imbalance had no impact for $SMD_{CS/BL}$ and $SMD_{CS/CS}$, but predicted higher divergences when using $SMD_{CS/EP}$. We did not find that study attrition (i.e., the proportion of patients lost to follow-up) had a significant influence on effect size differences, except when assuming $r = 0.8$ for $SMD_{CS/CS}$. Comprehensive results are tabulated in S3 Table.

## Discussion

In this study, we examined the impact of different SMD calculation methods on the meta-analytic effects of depression psychotherapy. We examined different ways to compute the unstandardized mean difference (endpoint scores versus change from baseline), and different standardizing denominators. For psychotherapy compared to inactive control groups, results differed substantially depending on the calculation method, with SMDs ranging from 0.65 to 1.23. Such differences can have a major impact on the clinical interpretation of treatment effects: following Cohen's "operational definition" [48], the lowest estimate would indicate a medium-sized effect of psychotherapy, while the highest estimate represents a very large effect; even though the same data was used. These variations present a considerable risk if meta-analytic results are naïvely compared to other reviews or trials using a different SMD calculation method. Our findings illustrate, as others have before [10,28], that SMDs can be much less "standardized" than their name suggests.

Our results co-align with statistical theory, which predicts divergences between SMD estimates depending on the denominator used to "standardize" the effect (see, e.g., results of our simulation in S1 Text, which closely mirror our empirical findings). Holding the raw mean difference constant, SMDs will increase as the SD in the denominator decreases, and this effect will be most pronounced when the raw mean difference is large. This will often be the case in trials comparing

effective treatments against "weak" comparators (e.g., waiting list, placebo, or other inactive controls [49]). This may also explain why divergences were considerably smaller for comparisons of psychotherapy to ADM ($SMD = 0.05$ to $0.14$), where only minor between-group differences are typically found. However, even in this dataset, SMD variants could lead to markedly dissimilar estimates of the heterogeneity variance. Standardization could also explain our observed difference between endpoint and change score SMDs when the latter are divided by the pre-test SD. Compared to post-test, pre-test SDs in RCTs may often be restricted due the application of cut-offs or floor effects, which leads to higher SMDs on average. Consistent with this, smaller divergences were found when change score SMDs were calculated using the post-test SD instead. When change score SDs were used in the denominator, we found a very strong dependence on the pre-post correlation; pooled effects compared to inactive controls differed by $\Delta_{SMD} = 0.58$ depending on whether high ($r = 0.8$) or low ($r = 0.2$) values were assumed.

This is problematic for aggregate-data meta-analyses. In-sample correlations will seldom be reported for every study, and therefore must be imputed using a sensible "guesstimate" from the literature. A previous review reported a median pre-post correlation of $r = 0.36$ for psychiatric interventions (25$^{th}$ percentile: 0.22; 75$^{th}$ percentile: 0.58), the lowest among all fields of medicine [36]. This value is close to $r = 0.4$, for which divergences between $SMD_{EP}$ and $SMD_{CS}$ in our analyses were smallest, at least when change score SDs were used. In meta-analyses where the correlation can be obtained from each study (or individual participant data is available), differences between $SMD_{EP/EP}$ and $SMD_{CS/CS}$ might therefore often be less pronounced. A recent investigation using IPD meta-analysis confirmed this [50], and further research may be helpful to corroborate this finding. Major discrepancies may still be possible if pre-post correlations vary considerably across trials, or if there are subfields with persistently higher or lower within-group correlations.

Table 2 presents practical recommendations for calculating SMDs using aggregate data from mental health trials. Overall, we believe our results seriously question the usefulness of change score SMDs – at least in meta-analytic psychotherapy research. Contrary to common belief, this calculation method does not adequately control for patients' baseline symptomatology in most cases; yet it creates considerable ambiguity as to what plug-in estimator should be used in the standardizing denominator. Some have proposed that the pre-test SD should be used for this purpose [21,22], but our findings indicate that $SMD_{CS/BL}$ can lead to substantially larger effect estimates than $SMD_{EP/EP}$. A recent simulation study suggested that change score SMDs may be less biased for studies with attrition at follow-up [22], but we did not find that this had a significant impact on the relationship between change score and endpoint SMDs; neither did the strength of baseline imbalance within studies. If change score SDs are used instead, the (pooled) SMD will strongly depend on how efficient change scores are as estimators of the true treatment effect compared to endpoint scores, and this is largely determined by the (true or "guesstimated") pre-post correlation we happen to find in a specific trial (cf. equation 11). Arguably, none of these are desirable properties for a "standardized" effect that should facilitate comparing results across trials, treatments, or research fields.

There is less ambiguity concerning the calculation of endpoint SMDs (viz., $SMD_{EP/EP}$). Also, should $r = 0.36$ hold as a generally representative value for psychiatric contexts, endpoint mean differences might come closer to ANCOVA-based estimates (cf. equation 10). Prioritizing endpoint SMDs in meta-analyses should also be practically feasible; for example, post-treatment means and SDs could be extracted from 88.4% (inactive controls) and 74.8% (ADM) of all trials included in the depression psychotherapy databases we analyzed here. In trials that only report the change from baseline, mean differences could also be standardized by the endpoint SD, since this led to only minor differences in our analysis. If not reported, there is empirical support for borrowing endpoint SDs from the other studies [51].

In our analysis, 16.7% to 27.3% of trials had to be removed because they did not report group means and standard deviations at both pre- and post-test. This indicates that, in general, outcome reporting in psychotherapy trials needs to be improved. To enhance transparency, researchers may also provide a pre-specification of the SMD calculation method they plan to employ, to prevent selective reporting of the one variant yielding the highest effect size. In this context, it is important to re-emphasize that none of the SMD calculation methods we examined here is inherently "wrong" or biased.

Researchers may still select a different variant than $SMD_{EP/EP}$ for their analysis; but this should be clearly described, since it could limit the comparability of the effect size. We also want to underline that our recommendations in this paper are purely pragmatic, and may not translate to every context in mental health research. Examples include quasi-experimental designs, or sub-fields in which pre-post correlations are typically reported.

Finally, one should not gloss over the fact that even endpoint SMDs use a "plug-in" estimate of the population SD, which will depend on the overall variability in the trial. SDs may still differ between tightly controlled studies and, say, pragmatic trials with broad inclusion criteria. It has been recommended that, instead of computing study-specific estimates, meta-analysts should employ external SD estimates, with the same reference value used for each scale ([52]; so a common SD for, e.g., the Patient Health Questionnaire, Hamilton Depression Rating Scale, Beck Depression Inventory, etc.). This could further improve the comparability and transportability of effect estimates beyond a single trial.

Naturally, the optimal solution would be if all measurements in RCTs were standardized to begin with. There are increasing efforts to establish core outcome sets (COS [53]) to be included in all clinical studies within a research field, including psychological treatment [54–56]. Most "perils of standardization" we examined in this paper could be avoided altogether if such standards were more widely adopted.

## Supporting information

**S1 Fig. Estimated relationship between true effect sizes for different calculation methods of the SMD.** $\Delta_{SMD}$ = Difference between the pooled effect based on endpoint SMDs ($SMD_{EP/EP}$), and SMDs calculated using change scores ($SMD_{CS}$). "Attrition" refers to the proportion of participants who were lost to follow-up, pooled across both trial arms (continuous covariate); "Baseline Imbalance" to the absolute value of the between-group SMD at baseline (continuous covariate); and "Risk of Bias" to the number of domains assessed to have a low risk of bias (continuous covariate; 0–4).
(TIFF)

**S1 Text. True effect and estimated SMD conditional on calculation methods (Simulated example).** No legend.
(PDF)

**S2 Text. References of the included studies.** No legend.
(PDF)

**S1 Table. Pooled effects of psychotherapy versus control groups, based on different calculation methods of the SMD.** $SMD_{EP/EP}$ = SMD calculated by dividing the mean endpoint difference by the pooled endpoint SD; $SMD_{CS/BL}$ = SMD calculated by dividing the mean change score difference by the pooled baseline SD; $SMD_{CS/CS}$ = SMD calculated by dividing the mean change score difference by the pooled change score SD; $SMD_{CS/EP}$ = SMD calculated by dividing the mean change score difference by the pooled endpoint SD.
(PDF)

**S2 Table. Pooled effects of psychotherapy versus pharmacotherapy trials, based on different calculation methods of the SMD.** $SMD_{EP/EP}$ = SMD calculated by dividing the mean endpoint difference by the pooled endpoint SD; $SMD_{CS/BL}$ = SMD calculated by dividing the mean change score difference by the pooled baseline SD; $SMD_{CS/CS}$ = SMD calculated by dividing the mean change score difference by the pooled change score SD; $SMD_{CS/EP}$ = SMD calculated by dividing the mean change score difference by the pooled endpoint SD.
(PDF)

**S3 Table. Divergent effect estimates, conditional on study and treatment characteristics (Psychotherapy versus Control).** $\Delta_{SMD}$ = Difference between the pooled effect based on endpoint SMDs ($SMD_{EP/EP}$), and SMDs calculated using change scores ($SMD_{CS}$). "Attrition" refers to the proportion of participants who were lost to follow-up, pooled across both

trial arms (continuous covariate); "Baseline Imbalance" to the absolute value of the between-group SMD at baseline (continuous covariate); and "Risk of Bias" to the number of domains assessed to have a low risk of bias (continuous covariate; 0–4).
(PDF)

## Author contributions

**Conceptualization:** Mathias Harrer, Clara Miguel, Edoardo G. Ostinelli, Eirini Karyotaki, Stefan Leucht, Toshi A. Furukawa, Pim Cuijpers.

**Data curation:** Mathias Harrer, Clara Miguel, Edoardo G. Ostinelli, Eirini Karyotaki, Pim Cuijpers.

**Formal analysis:** Mathias Harrer, Pim Cuijpers.

**Funding acquisition:** Pim Cuijpers.

**Investigation:** Mathias Harrer, Yan Luo, Edoardo G. Ostinelli, Toshi A. Furukawa, Pim Cuijpers.

**Methodology:** Mathias Harrer, Clara Miguel, Yan Luo, Edoardo G. Ostinelli, Toshi A. Furukawa, Pim Cuijpers.

**Project administration:** Mathias Harrer, Clara Miguel, Eirini Karyotaki, Pim Cuijpers.

**Resources:** Clara Miguel, Stefan Leucht, Pim Cuijpers.

**Software:** Mathias Harrer.

**Supervision:** Eirini Karyotaki, Stefan Leucht, Toshi A. Furukawa, Pim Cuijpers.

**Validation:** Mathias Harrer, Clara Miguel, Yan Luo, Edoardo G. Ostinelli, Eirini Karyotaki, Pim Cuijpers.

**Visualization:** Mathias Harrer.

**Writing – original draft:** Mathias Harrer, Clara Miguel, Yan Luo, Edoardo G. Ostinelli, Eirini Karyotaki, Toshi A. Furukawa, Pim Cuijpers.

**Writing – review & editing:** Mathias Harrer, Clara Miguel, Yan Luo, Edoardo G. Ostinelli, Eirini Karyotaki, Stefan Leucht, Toshi A. Furukawa, Pim Cuijpers.

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
