## [Decision Letter · Decision Letter 0]

PMEN-D-24-00574

Divergences Between Change Score and Endpoint Effect Sizes in Aggregate Data Meta-Analyses: A Primer and Empirical Study in Depression Psychotherapy

PLOS Mental Health

Dear Dr. Harrer,

Thank you for submitting your manuscript to PLOS Mental Health. After careful consideration, we feel that it has merit but does not fully meet PLOS Mental Health’s publication criteria as it currently stands. It is well written and is expected to meet criteria for publication, with revisions. Therefore, we invite you to submit a revised version of the manuscript that addresses the points raised during the review process.

Aditionally I have comments and suggestions:

1. Why not simulate data and then show what happens to the true effect under various scenarios, including missing data, intention to treat etc.? I agree with the reviewer on this point. I would have done the whole thing as a simulation and then used published studies as examples. I'm not suggesting to rework the paper entirely, but it would be useful to see at least a visual simulation study for comparison.

2. Add specific definition of 'generalisability' (P4, L87)

3. I do not understand why selection of baseline, change or endpoint scores is an issue specific to standardization - the same choice might be made for unstandardized scores. I may have misunderstood this, but then so might readers, so it should be rewritten here to ensure it cannot be misunderstood.

4. When discussing different instruments (e.g. P7L138 and elsewhere) please be clear if you are referring to depression specifically - lots of trials use instruments which are arguably 'common mental disorder' or 'psychological distress' (GHQ, HADS) or proxie for neuroticism, rather than depression specifically. You return to the issue of specific measures in the discussion, but it needs to be clear throughout and in relation to your selection criteria.

5. I am not an expert in RCT methods and was surprised/alarmed to see (P16L351) that sudies could be published without pre/post test means, SDs and sample sizes! Why were 532 - 443 studies even published? Is study quality really that permissably low, or has something happened with the inclusion criteria? Amend for clarity.

6. 'SMDs can be much less "standardized" than their name suggests: [other details]' would make a nice title for the paper, it conveys the crux of the message. If not, it should in my opinion be used in any summaries, press releases, dissemination activities etc.

7. Presumably some researchers will use different approaches to standardization and then publish the one giving the largest effect. Would pre-registration of the choice address this? Please add thoughts in relation to this (P21).

We look forward to receiving your revised manuscript.

Kind regards,

Gareth Hagger-Johnson

Academic Editor

PLOS Mental Health

Journal Requirement:

https://journals.plos.org/mentalhealth/s/figures 

https://journals.plos.org/mentalhealth/s/figures#loc-file-requirements 

Reviewers' comments:

Reviewer's Responses to Questions

**Comments to the Author**

1. Does this manuscript meet PLOS Mental Health’s publication criteria ? Is the manuscript technically sound, and do the data support the conclusions? The manuscript must describe methodologically and ethically rigorous research with conclusions that are appropriately drawn based on the data presented.

Reviewer #1: No

Reviewer #2: Yes

2. Has the statistical analysis been performed appropriately and rigorously?

Reviewer #1: Yes

Reviewer #2: Yes

3. Have the authors made all data underlying the findings in their manuscript fully available (please refer to the Data Availability Statement at the start of the manuscript PDF file)?

Reviewer #1: Yes

Reviewer #2: Yes

4. Is the manuscript presented in an intelligible fashion and written in standard English?

Reviewer #1: Yes

Reviewer #2: Yes

5. Review Comments to the Author

Reviewer #1: This work presents the results of a methodological review on the consequences of using different effect size indices from the 'standardized mean difference' family in meta-analyses on the efficacy of psychological interventions. Although the study focuses on research about the efficacy of psychological interventions for treating depression, its findings are generalizable to other meta-analyses on the efficacy of psychological interventions and may even be applicable to other types of interventions in the Social and Health Sciences. The essential requirement for this applicability is that the empirical studies feeding these meta-analyses must have employed a pretest-posttest design with a control group (either with or without random assignment, i.e., experimental and quasi-experimental studies). In such studies, the basic descriptive statistics needed to apply any version of the 'standardized mean difference' are the pretest and posttest means and SDs of the two groups being compared, the SDs of the change scores, and, if possible, the Pearson correlation coefficients between the pretest and posttest scores for the two groups.

The authors of this study have conducted a very comprehensive investigation by analyzing a large database of primary studies (particularly regarding comparisons between treated and control groups, with k = 395 studies and 791 effect sizes). In my opinion, publishing methodological reviews of this kind is highly important as they provide a realistic overview of the extent to which using different effect size indices can impact the results of a meta-analysis. However, I find certain aspects of the work that would require further theoretical and conceptual development. My interpretation of the results of this research does not align with the authors’ conclusions. Therefore, my recommendation is that the authors undertake a thorough revision of the aspects that I present below.

(1) In the Introduction of this paper, it should be made clear that different indices from the 'standardized mean difference' family estimate different parameters. These parameters differ both in the numerator and denominator of the effect size formula in question. In the numerator, there are essentially two options: the posttest mean difference and the difference between the pretest-posttest mean changes of the two groups (setting aside the use of adjusted posttest means for the pretest). In the denominator, there are different standardizers, each of which leads to estimating different parameters: SDEP, SDBL, and SDCS. Therefore, in the Introduction, each estimator should be linked to its corresponding parameter. Specifically, I am referring to the estimators SMDEP/EP, SMDCS/BL, SMDCS/CS, and SMDCS/EP.

(2) Using the posttest mean difference in the numerator implies disregarding the baseline levels of the two groups in the pretest, which could result in biased estimates of the true population effect. This is particularly important in quasi-experimental studies (i.e., without random assignment) but can also occur in experimental studies. From the perspective of controlling threats to the internal validity of research findings, using the difference between the pretest-posttest mean changes of the two groups is preferable to ignoring the pretest means (cf., e.g., Shadish et al., 2002). Therefore, conceptually, it is preferable to use the difference between the pretest-posttest mean changes in the numerator rather than the posttest mean difference. Based on this premise, the parameter we wish to estimate is this difference divided by a standardizer. If the parameter of interest is that difference, then it no longer makes sense to use SMDEP/EP as the "gold standard." The authors' conclusion that SMDCS/BL or SMDCS/EP overestimates the true population effect is based on treating SMDEP/EP as the ideal effect size, which is an unwarranted assumption. In other meta-analyses (outside the study of depression), it is possible that using SMDCS/BL or SMDCS/EP could result in lower estimates than SMDEP/EP. This is an empirical question that may vary across fields. However, the important point is that conclusions about the performance of SMDCS/BL or SMDCS/EP cannot be drawn based on SMDEP/EP as the "gold standard" if the meta-analyst is interested in estimating a parameter different from that estimated by SMDEP/EP. In my experience replicating meta-analyses, I have encountered examples where using SMDEP/EP yielded a highly significant pooled effect favoring the treatment, while SMDCS/BL produced a mean effect of zero.

(3) The use of SMDCS/BL or SMDCS/EP might be limited if the results varied drastically depending on the imputed correlation. However, the results presented in this study do not suggest this. In Table 1, we can see that the pooled effects of SMDCS/BL for correlations of 0.2, 0.4, 0.6, and 0.8 were 0.92, 0.92, 0.93, and 0.93, respectively. For SMDCS/EP, the mean effects were 0.82, 0.83, 0.83, and 0.83. The results presented in the supplementary tables are consistent with this. That is, the imputed correlation had a negligible effect on the pooled effect. Therefore, SMDCS/BL and SMDCS/EP are perfectly applicable estimators, even if the correlation needs to be imputed. Furthermore, conceptually, they are preferable to SMDEP/EP as they account for pretest scores.

(4) If the meta-analyst decides to use SMDCS/BL or SMDCS/EP, they can approximate a correlation by deriving it from studies in the meta-analysis that do report this correlation, averaging it, and imputing it to the rest of the studies that do not report it. This strategy would allow for a reasonably accurate approximation of the true pretest-posttest correlation. Note that some primary studies may not report the pretest-posttest correlation but do report descriptive statistics from which it can be calculated. For instance, if a study reports SDBL, SDEP, and SDCS, the correlation can be computed. If the study reports SDBL, SDEP, and the t-test comparing pretest and posttest means, the correlation can also be calculated. The authors should clarify these strategies to impute correlations as close as possible to the true parametric correlation.

(5) The decision of which effect size index to use in a meta-analysis must be based on the parameter the meta-analyst wishes to estimate. If the meta-analyst does not wish to account for pretest means, he/she should use SMDEP/EP. If the meta-analyst wants to consider the pretest-posttest mean changes of the two groups, he/she can use SMDCS/BL, SMDCS/EP, or SMDCS/CS. Among these options, I recommend always using the difference between the pretest-posttest mean changes. Once this decision is made, the next step is to choose the standardizer (i.e., the denominator). Since SMDCS/CS is heavily influenced by the imputation of the correlation coefficient, the decision would be between SMDCS/BL and SMDCS/EP. If the meta-analyst is interested in estimating the parametric effect relative to participants’ status after treatment, the estimator of choice should be SMDCS/EP. If the meta-analyst considers that SDBL might be a more homogeneous estimator of variability across studies (as SDBL is unaffected by the treatment effect), he/she may opt for SMDCS/BL.

In summary. Based on the results obtained by the authors, my conclusions differ significantly. The authors' preference for SMDEP/EP is neither empirically supported (SMDCS/BL and SMDCS/EP perform well) nor conceptually justified (using the difference between the pretest-posttest mean changes is conceptually superior to relying solely on posttest means).

Reference

Shadish, W. R., Cook, T. D., & Campbell, D. T. (2002). Experimental and quasi-experimental designs for generalized causal inference. Houghton, Mifflin and Company.

Reviewer #2: This manuscript tackles a longstanding and important problem in meta-analytic statistics for psychotherapy research: how to handle standardised mean differences (SMDs) when different approaches to calculating them can yield widely divergent summary estimates. The authors provide a thorough examination of change-score versus endpoint-based calculations, illustrating the practical implications through a large dataset on depression psychotherapy. Given the extensive reliance on SMDs in mental health research (and psychology research more broadly), the paper is both timely and methodologically instructive.

The manuscript is well-organised: it starts by clearly laying out theoretical relationships among the various versions of SMD, then demonstrates these relationships empirically using a rich database. I commend the authors’ careful approach to documenting and justifying each analytical decision. The open-data and open-code also enhances reproducibility.

The finding that psychotherapy vs. control effects varied from 0.65 to 1.24, depending on the SMD calculation, is a striking result. This convincingly demonstrates to readers that “standardisation” is far from consistent. The manuscript rightly cautions against naïve comparisons of meta-analytic results that use different SMD methods. The authors rightly acknowledge several limitations—chief among them the reliance on aggregate data. It is not obvious to me why the study was not conducted using a public IPD dataset or just simulated data. From a readers perspective doing so would have meant the authors could claim the effect sizes not just vary wildly, but that some effect sizes are *wrong* by X. By relying on aggregate data, it seems the authors are deprived of 'ground truth.' Using IPD or simulated data would mean the authors could compare the meta-analytics estimates against that ground truth, showing that their recommended methods are both pragmatic (e.g., they don't need each study to report a pre-post correlation coefficient) and accurate (i.e., the estimate closely reflects the true value).

The discussion section appropriately highlights the challenge of choosing an SMD. The authors ultimately suggest prioritising endpoint-based SMDs (SMDEP/EP) in meta-analyses and, when possible, using a common reference SD if outcomes are measured on the same scale. They also mention the possibility that future core outcome sets or standardised measurement approaches might do away with many of these complexities. In the meantime, I would encourage the authors to consider offering a pragmatic guideline or flowchart for researchers designing or synthesising new trials (a la https://bmcmedresmethodol.biomedcentral.com/articles/10.1186/s12874-017-0442-1). Such recommendations would be invaluable for future meta-analysts.

Some minor issues could likely be picked up in copy-editing and are of less substantive concern. For example, I assume IPD refers to 'individual patient data' but many psychotherapy researchers might not know this acronym; likewise clarify beta in equation 2.

Otherwise, this paper systematically quantifies the discrepancies in meta-analytic effect sizes that stem from different SMD calculation methods. It is of considerable importance for any researchers synthesising trial evidence on psychological interventions with heterogenous outcomes.

6. PLOS authors have the option to publish the peer review history of their article (what does this mean? ). If published, this will include your full peer review and any attached files.

**Do you want your identity to be public for this peer review?** For information about this choice, including consent withdrawal, please see our Privacy Policy .

Reviewer #1: No

Reviewer #2: **Yes: ** Michael Noetel

---

## [Decision Letter · Decision Letter 1]

Standardized Effect Sizes Are Far From "Standardized": A Primer and Empirical Illustration in Depression Psychotherapy Meta-Analyses

PMEN-D-24-00574R1

Dear Mr Harrer,

We are pleased to inform you that your manuscript 'Standardized Effect Sizes Are Far From "Standardized": A Primer and Empirical Illustration in Depression Psychotherapy Meta-Analyses' has been provisionally accepted for publication in PLOS Mental Health.

Thank you for a conscientious revision, with clearly documented responses to reviewers and editorial requests.

Best regards,

Gareth Hagger-Johnson

Academic Editor

PLOS Mental Health

**Comments to the Author**

1. If the authors have adequately addressed your comments raised in a previous round of review and you feel that this manuscript is now acceptable for publication, you may indicate that here to bypass the “Comments to the Author” section, enter your conflict of interest statement in the “Confidential to Editor” section, and submit your "Accept" recommendation.

Reviewer #2: All comments have been addressed

2. Does this manuscript meet PLOS Mental Health’s publication criteria ? Is the manuscript technically sound, and do the data support the conclusions? The manuscript must describe methodologically and ethically rigorous research with conclusions that are appropriately drawn based on the data presented.

Reviewer #2: Yes

3. Has the statistical analysis been performed appropriately and rigorously?

Reviewer #2: Yes

4. Have the authors made all data underlying the findings in their manuscript fully available (please refer to the Data Availability Statement at the start of the manuscript PDF file)?

Reviewer #2: Yes

5. Is the manuscript presented in an intelligible fashion and written in standard English?

Reviewer #2: Yes

6. Review Comments to the Author

Reviewer #2: The authors have made substantial amendments to the manuscript in response to our (the reviewers’ and editor’s) comments, including clarifications of their methodological rationale, the addition of a supplementary simulation, and the provision of practical recommendations for standardised mean difference (SMD) calculations. These enhancements ensure that the paper offers a more precise understanding of endpoint- vs. change-score-based SMDs and how different calculation methods can influence meta-analytic outcomes.

7. PLOS authors have the option to publish the peer review history of their article (what does this mean? ). If published, this will include your full peer review and any attached files.

**Do you want your identity to be public for this peer review?** For information about this choice, including consent withdrawal, please see our Privacy Policy .

Reviewer #2: **Yes: ** Michael Noetel
